# Molecular Characterization of Small Ruminant Lentiviruses in Sheep and Goats: A Systematic Review

**DOI:** 10.3390/ani14233545

**Published:** 2024-12-08

**Authors:** Paola Gobbi, Silvia Pavone, Massimiliano Orso, Fabrizio Passamonti, Cecilia Righi, Maria Serena Beato, Francesco Feliziani, Monica Giammarioli

**Affiliations:** 1National Reference Laboratory for Ruminant Retroviruses, Istituto Zooprofilattico Sperimentale dell’Umbria e delle Marche “Togo Rosati” (IZSUM), Via G. Salvemini 1, 06126 Perugia, Italy; s.pavone@izsum.it (S.P.); c.righi@izsum.it (C.R.); ms.beato@izsum.it (M.S.B.); f.feliziani@izsum.it (F.F.); m.giammarioli@izsum.it (M.G.); 2Department of Veterinary Medicine, University of Perugia, Via San Costanzo 4, 06126 Perugia, Italy; fabrizio.passamonti@unipg.it; 3Office for Research Management, Special Projects, Cooperation and Twinning, Istituto Zooprofilattico Sperimentale dell’Umbria e delle Marche “Togo Rosati” (IZSUM), Via G. Salvemini 1, 06126 Perugia, Italy; m.orso@izsum.it

**Keywords:** maedi-visna, caev, small ruminant lentivirus, SRLVs, sheep, goats, small ruminants, phylogenesis, molecular characterization, genotyping

## Abstract

Small ruminant lentiviruses (SRLVs) are responsible for a disease complex that includes a variety of clinical forms with a large degree of severity. The virus is highly variable, and 5 genotypes with 34 subgenotypes have been described so far. However, the application of different protocols for genotyping generated contradictory results with potential misclassification of some strains and/or identification of redundant new subgenotypes. To the best of our knowledge, no systematic review on the molecular characterization of SRLVs in sheep and goats is available. The present systematic review aims to provide an updated, in-depth, comprehensive overview of the phylogenesis of SRLVs. The systematic review was developed according to the PRISMA-P statement.

## 1. Introduction

Small ruminant lentiviruses (SRLVs) are single-stranded RNA viruses grouped in the order *Ortervirales*, *Retroviridae* family, *Orthoretrovirinae* subfamily, and *Lentivirus* genus [1,2]. The SRLV genome consists of two linear molecules of single-stranded positive RNA that are converted to double-stranded (ds) DNA via the viral enzyme reverse transcriptase (RT), and then the viral genome is integrated into the host genome as a provirus [3]. For a long time, the two members of this group, caprine arthritis–encephalitis virus (CAEV) and maedi-visna virus (MVV), have been considered two different strictly host-specific viruses of the genus, which infect goats and sheep, respectively [4,5,6,7]. MVV was the first lentivirus to be discovered and was isolated from sheep in Iceland in the 1950s [8]. Twenty years later, CAEV was isolated from goats [9]. To date, these viruses are no longer considered species-specific pathogens since they efficiently cross the species barrier between goats and sheep [10] and they are therefore grouped together as SRLVs [11]. SRLVs are currently categorized into five phylogenetic groups, from A to E. Genotype A in sheep consists of MVV-like strains, genotype B corresponds to CAEV-like isolates, and the remaining three groups include genotypes from specific geographical regions [12]. SRLVs are widespread worldwide, with Europe exhibiting the highest individual prevalence in sheep [13]. Seroprevalence rates vary across European countries, with Belgium and Switzerland reporting lower rates (13% in goats and 17% in sheep in Belgium; 9% in goats in Switzerland) compared to Poland and Spain, which recorded higher rates (72% in sheep in Poland; 53% in sheep in Spain) [14,15,16,17]. Low seroprevalence values may be due to national control and eradication plans [18], or to small herd sizes [14]. A few control and eradication plans have been implemented in Italy, though limited to the north-east region [19]. Several studies have shown a drastic reduction in seroprevalence values associated with CAEV, from 32% to less than 2% [19,20]. In southern Italy, seroprevalence remains high, reaching herd levels greater than 50% [21]. Intermediate prevalence rates are observed in Africa, Asia, and North America, while South and Central America appear to be less affected [22].

SRLVs cause multisystemic chronic and progressive disease characterized by inflammatory lesions in various organs, including lungs, brain, mammary glands, and joints. Sheep appear the most susceptible species, with manifestations including progressive interstitial pneumonia (maedi) leading to dyspnea and weight loss, demyelinating leukoencephalomyelitis (visna) resulting in neurological signs, indurative mastitis, and arthritis [23,24,25,26]. Conversely, infected young goats typically exhibit leukoencephalitis, while chronic arthritis is more common in adults (CAEV) [27]. These manifestations result in prolonged illness and reduce productivity, exerting a significant economic impact on the small ruminants industry [28].

SRLVs primarily target cells of the monocyte/macrophage lineage and dendritic cells, although other cell types can also be infected. Endothelial cells, mature and immature luminal epithelial cells, fibroblasts, and myoepithelial cells from goat mammary gland [29], as well as interstitial fibroblasts, acinar epithelial cells, macrophages, endothelial cells, and adipocytes of ovine mammary tissue [30], are susceptible to SRLV infection. This suggests that mammary epithelial cells play a key role in SRLV pathogenesis [31].

Colostrum and milk are recognized as the primary routes of SRLV transmission [32]. Consequently, control measures and eradication programs primarily focus on separating offspring from their mothers at birth and feeding them with colostrum and milk obtained from SRLV-free animals [33,34,35]. Other less efficient routes of transmission include prolonged close contact with infected animals, particularly in cases of the maedi form of MVV infection, where respiratory exudates may facilitate virus transmission. [36]. This mode of transmission is especially significant in intensive housing or grazing conditions where high animal density facilitates the spread of SRLVs [32]. Additionally, intrauterine and seminal routes are speculated to be potential sources of viral dissemination, although further investigations are required to confirm their role [37].

No gold standard test for SRLV diagnosis has been defined yet [38,39]. Serological techniques, such as agar gel immunodiffusion tests (AGID), Heteroduplex Mobility Assay (HMA), and enzyme-linked immunosorbent assays (ELISAs), along with molecular techniques like PCR and RT-PCR, are commonly used for diagnosis [38,40]. Nowadays, AGID has been replaced by ELISA due to its higher sensitivity, cost-effectiveness, ease of use, and quicker results. The World Organisation for Animal Health (WOAH) now recognizes ELISA as the recommended test for international trade. However, the antigenic diversity of SRLVs can sometimes evade detection using current monovalent serological tests [41]. Molecular tests, especially PCR and RT-PCR, are valuable for early detection of infection, particularly before seroconversion, and serve as a complement to serological tests [7,35]. However, PCR sensitivity may be reduced in animals with latent infection due to low viral load and high viral genetic heterogeneity [42]. Therefore, a combination of different laboratory tests, such as serology and PCR, is often recommended to improve the detection of infection [43,44].

The economic losses resulting from SRLV infections, including reductions in birth and growth weight, milk production, and premature culling, have underscored the significance of this disease, prompting its inclusion in the list of notifiable terrestrial animal diseases by the WOAH. Indeed, WOAH has included SRLVs in the list of notifiable terrestrial animal diseases highlighting the recommendations for importing sheep and goats [41]. Consequently, trade restrictions further exacerbate the economic impact of SRLVs. Nevertheless, Regulation (EU) 2016/429 on transmissible animal diseases [45] and the Commission implementing Regulation (EU) 2018/1882 [46] did not designate MVV and CAEV as diseases requiring the application of prevention and control rules. As a result, official health authorities across European countries have not mandated SRLV surveillance, control, or eradication programs, and all measures applied in each European country have been initiated voluntarily. Finland, Italy, Spain, France, Germany, the Netherlands, and Switzerland have applied voluntary control and eradication programs overtime [19,47,48,49,50]. However, the effectiveness of these programs in controlling the disease has been questionable [49].

The genome of SRLVs consists of three structural genes encoding group-specific antigens (*gag*), the polymerase (*pol*) and envelope (*env*) genes, along with regulatory genes including *vpr-like* (formerly *tat*), *rev*, and *vif*, which have information for the synthesis of proteins that regulate viral replication [51,52]. A schematic representation is shown in Figure 1. The *gag* gene encodes internal structural proteins, including matrix (p16MA), capsid (p25CA), and nucleocapsid (p14NC) proteins. The *pol* gene encodes enzymes such as reverse transcriptase (RT), protease (PR), and integrase (IN) enzymes involved in replication and DNA integration, with variations linked to SRLV pathogenicity. Indeed, the dUTPase subunit, encoded by the *pol* gene, has been found to be dispensable for viral replication [53]; however, dUTPase-negative strains produce less severe lesions, which are restricted to the injection site [54].

The *env* gene encodes transmembrane (gp46TM) and surface (gp135SU) glycoproteins involved in SRLV tropism (cell, tissue, and host species) [12,23]. Within the most variable portion of the *env* gene, recurring sequences, referred to as “signature patterns,” have been identified [55]. A correlation between these patterns and the pathogenicity of SRLV could exist, but more studies are needed to corroborate this hypothesis. The *vpr-like* gene enhances the viral load, tissue distribution, and inflammatory lesion severity over that of the *vpr-like* deletion counterpart [56]. The proviral DNA is flanked by non-coding sequences known as long terminal repeats (LTRs), subdivided into U3, R, and U5 regions, containing regulatory sequences crucial for viral replication and viral gene expression [57]. Additionally, LTRs play a role in viral tropism within the central nervous system, with increased neurovirulence observed when specific CAAAT sequences are duplicated [58]. Deletions or mutations in LTRs may be associated with reduction in virulence, likely due to the presence of replication enhancer elements such as AP1, AML, tumor necrosis factor-α, and gamma interferon response elements [58,59].

As it is well known, the *env* gene and LTRs exhibit a high level of variability, while the *gag* and *pol* genes are relatively conserved [10]. Initially, *gag* and *pol* were the regions mainly utilized for phylogenetic analyses of SRLVs due to the need for a reliable method to detect the high number of strains [60,61,62,63,64]. However, since the phylogenetic analysis of variable regions has proven to be more informative than that of conservative regions [65], and LTRs seem to have the highest level of phylogenetic information for SRLVs [66], recent studies have shifted their focus to the LTRs region [10,67]. In Poland, a comparison of *gag-env* and LTRs sequences was conducted, revealing a broad consensus in SRLV phylogenies. The phylogenetic trees obtained showed a nearly superimposed topology, suggesting that these genomic fragments have likely co-evolved. The minor discrepancies observed in affiliation were attributed to mixed flocks where more than one SRLV genotype circulated. These results confirm that mixed flocks facilitate the emergence of new SRLV variants after cross-species transmission, which subsequently evolve to adapt to a new host [10].

Several factors contribute to the formation of new subtypes in SRLVs. Most of the mutations in the SRLV genome are caused by the low fidelity of reverse transcriptase (RT) [12]. Additionally, macrophages, which are the main target for SRLVs, have an imbalanced level of deoxyribonucleotides (dNTPs) with an excess of dUTP, which can be incorporated into the viral genome due to the inability of RT to distinguish dTTP and dUTP [68]. These mutations lead to the creation of a population of distinct but genetically related viral variants known as quasispecies, which are found in infected animals [69]. The pivotal role of quasispecies in the phenomenon of compartmentalization is well established [60]. Given the ability of viruses to replicate across diverse anatomical environments or tissues, distinct evolutionary trajectories unfold, culminating in the emergence of compartment-specific quasispecies [70]. Notably, investigations have elucidated that predominant viral variants identified in the colostrum of goats [71], as well as in the central nervous system, lungs, and mammary glands of sheep exhibiting clinical manifestations of MVV, represent minor variants within the peripheral blood, as corroborated by *env* gene sequencing [70]. These quasispecies exhibit significant genetic and phenotypic divergence, shaped by the selective pressures inherent to their respective niches. As such, the compartmentalization of viral quasispecies reflects the virus’s adaptive response to the unique microenvironmental conditions within the host organism, exerting influence over various facets of pathogenesis, including viral transmission and immune modulation. Genetic drift and recombination in host cells during co-infections are two further significant events that contribute to the genetic diversity of SRLVs [72]. Recombination, especially in the *env* gene, is common and leads to a high incidence of antigenic variation, as well as changes in cellular host range, infectivity, cytopathogenicity, and disease progression [73]. Moreover, interactions with host factors such as molecules of innate and adaptive immunity like the APOBEC3 enzyme can induce deleterious mutation of the viral genome [12,62,74]. These combined factors result in the high genetic diversity of SRLVs, which poses challenges for the reliability of diagnostic techniques.

Since the 1990s, studies on molecular characterization of circulating SRLVs have increased, leading to the development of new and updated molecular tools for diagnosing such infections. PCR for proviral DNA [75,76,77,78,79,80,81] or RT-PCR for viral RNA [82] has often been used, but the diagnostic sensitivity of these methods appears to be hampered by the high degree of sequence diversity and low copy numbers [66,82,83,84]. Until 2004, SRLV phylogenetic studies were based on relatively short sequences within *gag*, *pol*, or *env* [65,66]. However, these short conserved sequences did not seem optimal for identifying the high genetic variability of SRLVs. In 2004, Shah et al. [60] suggested a new molecular approach based on longer sequences: 1.8 kilobase (kb) *gag-pol* or 1.2 kb long conserved *pol* sequences suitable for sensitive molecular diagnosis and phylogenetic analysis at the same time. Nevertheless, the high genetic variability of SRLVs can hamper the detection of these fragments, and classification cannot be consistently achieved. Therefore, more recent studies have focused on LTR sequences [10] or Next-Generation Sequencing (NGS) technology [85,86]. The use of all these different molecular approaches over time has led to a new classification that currently categorizes SRLVs into five genotypes (A–E) and at least 34 subgenotypes (A1–A27, B1–B5, and E1–E2) [10]. However, the affiliation of some strains is inconstant and depends on the genomic region selected for genotyping and subgenotyping, sequencing method, and phylogenetic analysis [62].

In the absence of a treatment or vaccine for SRLVs, the only means of containing their spread and preventing contagion is to apply biosecurity measures and to gain a thorough understanding of the epidemiological situation in specific geographical regions. This can be achieved using diagnostic tests that are specific, sensitive, and accurate. While numerous diagnostic techniques are currently available, not all are capable of providing reliable results promptly.

To date, no systematic review has been conducted to consolidate information on the molecular characterization of SRLVs in sheep and goats. Given the growth of genetic analysis in research and diagnostics, a systematic review addressing this topic is warranted. The aims of this systematic review are as follows:Critically analyze the existing literature on the molecular characterizations of SRLVs in sheep and goats.Detail the various molecular approaches currently in use for SRLV analysis.Identify gaps in knowledge that need to be addressed for a better understanding of SRLVs in sheep and goats.Suggest practical recommendations for routine molecular diagnostics and research applications to ensure robust and reliable results.

This review aims to serve as an important source of information for researchers by guiding the selection of optimal methodologies for SRLV molecular characterization when planning new phylogenetic studies, enhancing the robustness of phylogenetic analysis and reducing as much ambiguity as possible in subgenotyping.

## 2. Materials and Methods

### 2.1. Scoping Study and Systematic Review Registration

The scope of this review was to investigate the molecular characterization of SRLVs in sheep and goats, across different laboratories worldwide over time, critically summarizing available data to provide an updated, in-depth, comprehensive overview of SRLV phylogenesis. This review is based on a study protocol registered in Protocols.io available at https://www.protocols.io/view/phylogenesis-of-small-ruminant-lentiviruses-a-syst-n2bvj8zzwgk5/v2 (accessed on 1 August 2024) DOI: https://doi.org/10.17504/protocols.io.n2bvj8zzwgk5/v2 (accessed on 1 August 2024), which follows the Preferred Reporting Items for Systematic Reviews and Meta-Analysis (PRISMA-P) guidelines [87,88].

### 2.2. Literature Search Strategy

A comprehensive computerized search on the Excerpta Medica Database (Embase Classic), MEDLINE (via PubMed), Scopus, and Web of Science was conducted to identify published peer-reviewed articles written in English, without a publication date limitation. The search was performed in two steps, the first on 25 January 2023, and the second on 25 January 2024. The utilized blocks of key terms are available in the published protocol https://doi.org/10.17504/protocols.io.n2bvj8zzwgk5/v2 (accessed on 1 August 2024) and in Appendix A. Key terms were combined using Boolean operators and MESH and Emtree terms specific to each database [89]. When full texts were unavailable, the corresponding authors were contacted to obtain the missing data.

### 2.3. Selection Process and Inclusion/Exclusion Criteria

The data found in each database were recorded and maintained in the software EndNote™(v. X7.8), where duplicates were easily identified and removed. Then, reference screening was performed by two independent review authors (SP and PG) in two steps. Firstly, the reviewers screened the titles and abstracts, removing studies irrelevant to the review’s scope. Studies published in peer-reviewed journals describing the molecular characterization of SRLVs in sheep and goats were included. Wild animals were not considered in this systematic review. Subsequently, the selected full-text articles were screened for eligibility as indicated on the PRISMA flow diagram [90]; we removed studies that did not focus on molecular characterization. Any disagreements between the reviewers were resolved by consensus or by the decision of other review authors (MO, MG). Regarding the study design, cross-sectional studies, reviews, and case reports were included, while experimental or intervention studies, editorials, commentaries, dissertations, notes, conference abstracts, and conference proceedings were excluded. The reasons for exclusion were recorded for each reference screened.

### 2.4. Quality Assessment and Risk of Bias

The methodological quality of the included studies was assessed using the checklist developed by Moola et al. (2020) and modified for our purposes, as detailed in the protocol published on protocols.io, https://www.protocols.io/view/phylogenesis-of-small-ruminant-lentiviruses-a-syst-n2bvj8zzwgk5/v2 (accessed on 1 August 2024) [91]. For each checklist item, the percentage of “yes”, “no”, “unclear”, and “not applicable” answers was reported. Studies with ≥60% of applicable items achieved were considered satisfactory. Studies with 60% to 80% of applicable items achieved were considered good, while those with 81% to 100% were considered very good. The quality assessment of narrative review articles was performed with the SANRA tool [92]. Two review authors (SP, PG) independently assessed each study’s methodological quality, and any disagreement was resolved by two other authors (MO, MG).

### 2.5. Data Extraction

The information collected included the identified subgenotypes, accession numbers, targeted genomic regions, genomic region width measured in base pair (bp), molecular method (conventional PCR or real-time PCR) used for classification, sample type investigated, species (sheep or goat) in which the subgenotype was detected, year of publication, country of origin, references, and DOI. All data were organized in an Excel document (Microsoft Office v. 2013, Microsoft, Redmond, WA, USA). A schematic illustration of the Materials and Methods is presented in Figure 2.

## 3. Results

### 3.1. Search and Quality Assessment Results

The literature search process is summarized in Figure 3 (PRISMA 2020 flow diagram).

Overall, 1325 published articles were retrieved from the four databases. Only papers in English were considered. After removing 630 duplicates, the remaining 695 articles were considered suitable for selection based on their title and abstract. After this screening, 566 articles were excluded for the following reasons: 553 were irrelevant to the scope of this study, 11 were not peer-reviewed publications, and 2 were not written in English. Therefore, 129 articles were assessed for eligibility, of which 23 were excluded. A list of the 23 excluded articles and the reasons for exclusion are reported in Appendix A. Finally, 106 articles are considered in the present systematic review on the phylogenetic analysis of SRLVs.

Among the 106 articles, there were no systematic reviews, 2 were reviews, 3 were case reports, and the remaining 101 were cross-sectional studies. The evaluation of articles in percentages is briefly shown in Table 1. The evaluations of the cross-sectional studies and case reports can be found, respectively, in Appendix A.

The evaluation of narrative reviews is shown in Table 2, while an accurate assessment can be found in Appendix A.

### 3.2. Geographic Distribution of Studies, Animal Species, Samples Investigated, and Routes of Transmission

Among all the selected papers subjected to full assessment, one was conducted in Africa (specifically in Sudan), eight were conducted in Asia (China, Iraq, Japan, Philippines, and South Korea), sixty-six in Europe (Belgium, Czech Republic, Finland, France, Germany, Greece, Italy, Norway, Poland, Portugal, Slovenia, Spain, Switzerland, and Turkey), 22 in North America (Canada, Mexico, and USA), and nine in South America (Argentina and Brazil). No study was retrieved from Oceania.

Regarding the origin of the samples, 36 articles examined samples of goat origin, 31 articles were based on sheep samples, 35 articles examined both goat and sheep samples, and in 4 papers, the species was not specified.

The types of samples analyzed are shown in Figure 4. Peripheral blood mononuclear cells (PBMCs) were the most frequently analyzed matrix [23,60,94,95,96,97,98,99,100,101,102,103,104,105,106], followed by peripheral blood leukocytes (PBLs) [55,67,72,107,108,109,110,111,112,113,114,115,116], cell cultures supernatants (CCSs) [64,117,118,119,120,121,122,123,124,125,126,127], and blood [24,40,63,128,129,130,131,132,133,134]. However, the majority of studies used multiple types of samples (MTs), such as alveolar macrophages, bronchoalveolar lavage, colostrum, milk, spleen, and synovial fluid [3,39,71,83,85,135,136,137,138,139,140,141,142,143,144,145,146,147].

In this review, we found that fomites have been recognized as a potential route of contamination [128,132,148]. Specifically, Cardinaux and colleagues hypothesized and demonstrated the role of fomites through the partial sequencing of viruses isolated from a mixed flock where goats and sheep were kept strictly separated, as evidenced by their analysis of the hypervariable sequence SU4 [148]. A similar situation was identified also by Fras and colleagues analyzing the *gag* region [128].

### 3.3. Genomic Region Selected for Genotyping/Subgenotyping and Software Analysis

Generally, the genomic region selected as the target coincides with the region used for subgenotyping, with only three exceptions identified. In the case of Colitti et al., the authors genotyped with NGS and subgenotyped on the gag gene [85], while De Martin et al. genotyped with gag, pol, and env, focusing on gag and pol for subgenotyping [122]. Molaee et al. genotyped the gag-pol region and derived subgenotypes by analyzing gag sequences only [63]. Out of 106 articles, 45 used molecular protocols adapted to single genes; specifically, 25 articles focused on the gag gene [24,94,99,102,103,106,107,109,110,111,113,114,123,127,128,131,134,141,149,150,151,152,153,154,155], 9 articles focused on the env gene [55,71,73,101,129,132,136,145,156], 4 articles focused on the pol gene [39,97,104,135], 7 on the LTR gene [10,57,67,133,157,158,159], 7 adapted the Shah protocol [60] on nearly complete pol and gag-pol regions [62,96,98,112,137,142,160], and 2 adapted L’Homme protocol [150] on a complete gag gene [153,160]. Moreover, seven articles focused on the gag-pol region [61,63,64,122,143,147,161]. Considering multiplex protocols, forty-three articles developed or implemented molecular procedures that recognized multiple genes [3,23,26,40,42,60,62,65,66,72,83,85,95,96,98,105,108,112,115,116,118,119,121,124,125,130,137,138,139,140,142,144,148,160,162,163,164,165,166,167,168,169,170], and eight applied a combination of a partial and complete sequencing approach [23,54,96,100,117,120,125,142]. Finally, eight articles adopted an NGS approach [85,86,138,146,168,170,171,172]. Three articles did not specify the genomic region used or were unclassifiable [12,93,173].

Another evaluation criterion was the size of the genomic region used for the phylogenetic characterization. Figure 5 reports the length of the genomic regions used for subgenotyping. Some studies used multiple genomic regions or generated sequences of different amplitudes [42,83,98,124,139,140,144,150]. The shortest sequence detected was 129 bp [106], while the majority of papers focused on genomic regions of 2000 bp or longer [23,54,65,86,96,100,117,120,124,125,136,146,162,167,168,170,171,172]. Finally, 13 records did not specify the length of the sequenced region [12,66,72,93,119,122,133,142,148,157,165,169,173].

The different softwares used by the selected articles for phylogenetic analysis are represented in Figure 6.

MEGA software was the most frequently used for phylogenetic analyses, often employed alone or in combination with other programs [3,10,39,40,42,61,62,63,64,66,71,72,94,95,97,100,103,104,105,107,108,109,112,113,115,116,118,123,124,125,127,128,130,131,132,133,137,138,139,140,143,144,145,147,148,149,150,151,152,153,154,158,160,162,163,164,166,170,173]. More information on the programs and algorithms used are reported in Appendix A.

A review of the literature revealed that the choice of a specific gene for molecular characterization and analyses using different softwares can influence the results obtained, even when working on the same sample. Specifically, in the identification of genotype C, Michiels and colleagues focused on *gag-pol* and *pol* regions as indicated by Shah [60], and then analyzed the obtained sequences with Clustal W (included in the MEGA 7.0 software). The alignments were carried out with the NJ method with the ML replacement model and statistical confidence of 1000 replicates [62]. They confirmed their findings by using both the ML method and the Bayesian method.

In contrast, Olech and colleagues sequenced the *gag* and *env* genes, carrying out phylogenetic analyses using Geneious Pro 5.3. Phylogenetic trees were constructed using the Bayesian interference method based on the GTR replacement model, while pairwise genetic distance was calculated with MEGA 6 software using the p-distance substitution model [107]. Finally, Kuhar and colleagues focused on the *gag* gene, building on previous works [76,94], and performed the phylogenetic analysis using Clustal W (included in the MEGA 4 software). They also worked with larger sequences, constructing two types of phylogenetic trees: one using the NJ method with MEGA and the other with the ML method using Phyml 3, both based on 1000 replicates. The genetic distances were calculated using the Tamura–Nei substitution model [112].

### 3.4. Strategic Challenges to SRLV Diagnosis: Operational Applications and Research Activities

Due to the high variability of SRLVs and the lack of an analytical gold standard, serological screening remains essential for accurate and reliable diagnosis. Consequently, a combined serological/molecular approach is often used by researchers in operational applications in diagnostic activity. Specifically, some authors observed varying degrees of discrepancies when comparing serological results with real-time PCR results [67,110,113,148,149,162,164]. Conversely, other authors achieved serological results that were in agreement with molecular results, in some cases nearly 100% concordant [85,108,122,144,154]. Interestingly, Schaer et al. [166] compared several serological tests and serological tests with real-time PCR, finding high variability among the former, but a certain concordance between molecular and serological results. Finally, some authors used a combined approach to improve diagnostic efficiency [72,95,111,127,128,132,139,140,141,155] or to develop specific tests for genotype E [54,155]. However, the researchers did not report a thorough comparison of the results from the serological and molecular tests.

### 3.5. Global Distribution of SRLV Subgenotypes and Phylogenetic Analysis

The global distribution of SRLVs subgenotypes obtained by the retrieved papers is shown in Figure 7 (source https://www.rawgraphs.io, accessed on 06 December 2024). This representation must be considered partial because some papers do not clearly correlate the subgenotypes of sequenced samples to accession numbers or to specific labels in the phylogenetic trees, making it impossible to consider their results in those cases. Genotypes A and B account for the majority of identified sequences. Italy provided the most sequences, followed by Poland. Outside Europe, robust epidemiological data are available for Brazil, the USA, Mexico, China, Japan, Sudan, South Korea, the Philippines, and Argentina, as shown in Figure 7. Samples belonging to genotypes C-D-E, often also referred to as groups, have recently been identified and appear to be related to well-defined geographical areas. Specifically, group C was found only in Norway [124,125], group D in Switzerland and Spain [60,98], and group E in several Italian regions [141,147,161,174], the latter being characterized by high genetic divergence and low pathogenicity [54,147,161].

Initially, SRLV infections were considered host-specific [7,8,9,175]. However, it has been clearly demonstrated that cross-species transmission occurs [97,98,118,119,138,140,142], although it is not entirely excluded that some strains can better adapt to a specific host [98]. Genotype E seems to be an exception, as to date, it has been isolated exclusively from goat samples [54,141,147]. Since its first isolation, its distribution in Italy has not seemed widespread but instead linked to specific regions [141,147,161]. Several studies have been carried out in Sardinia and Piedmont, identifying 84% similarity between Sardinian and Piedmontese isolates. This value makes the isolates similar enough to be included in the same genotype but different enough to be included in distinct clusters [141]. This clustering was later confirmed [147]. Geographical distribution appears to play a significant role in the evolution of SRLVs and the emergence of new strains [128].

The first attempt at a comprehensive philogeny of SRLVs was made by Zanoni in 1998 [66]. In this study, all available DNA sequences of SRLVs up to 1998 were retrieved from European Molecular Biology Laboratory (EMBL) nucleotide sequence databases, with data grouped by genome localization: LTR, gag, pol, and env sequences. Several tree-building methods were adopted and the results were compared, identifying six different clades (specifically Felsenstein’s F84 model, maximum likelihood [ML], neighbor-joining [NJ], and minimum evolution method). In 2004, Zanoni’s study was resumed [66] and updated by Shah and colleagues, who focused on a 1.8 kb gag-pol fragment and 1.2 kb pol fragment [60]. Shah’s main objective was to select one or more genomic regions of sufficient size to effectively represent the subgenotypes circulating in Switzerland. The gag and pol regions were selected for being highly conserved and considered suitable for developing sensitive molecular analysis protocols. Initially, all samples were screened with a short gag protocol (208 bp) to streamline the procedure. All positive samples were tested with the 1.8 kb gag-pol and the 1.2 kb pol procedures. Interestingly, they analyzed all sequences using different tree-building approaches (distance-based F84 substitution model and parsimony), obtaining comparable results. This demonstrated the robustness of the method and the validity of the data collected. By organizing the data from the different trees into boxplots to provide a more immediate representation of pairwise genetic distances, Shah and colleagues define statistical values for classifying samples into specific groups. Distances greater than 25% were observed when comparing samples from different genotypes; distances within the range 17–24% for the gag-pol region and the range 15–26% for pol were found when comparing samples of the same genotype but different subgenotypes. Differences of less than 15% for both regions were attributed to subtypes within the same subgenotype [60].

### 3.6. Advanced Research Applications: NGS, Recombination Events, and Compartmentalization

Molecular characterization is also a valuable tool in more sophisticated fields of research. The use of NGS techniques in SRLV studies remains limited, with only a few studies using this approach. Three papers were retrieved from USA [146,171,172], two from Europe [85,138], and three from China [86,168,170]. Workman and colleagues were the first to apply this method, defining the genotype A2 subgroup 4, while Colitti and colleagues identified the presence of two novel subgenotypes, A18 and A19, and confirmed the high genetic variability of Italian SRLVs [85]. Using the NGS approach, Olech and colleagues conducted an in-depth study of SRLVs at the single-nucleotide variation (SNV) level and demonstrated the existence of a quasispecies variant of SRLV [138]. One study originating from China identified the subgenotype A2 [86], while Wu and colleagues highlighted common ancestors with other Chinese strains [168]. Finally, Wang and colleagues identified the subgenotype B1, finding a correlation with the geographical distribution of seven other Chinese CAEV strains [170].

From the analysis of the selected papers, the study of recombination events is still relatively rare. Moreover, recombination events are not necessarily found in all tested samples, even when searched [61,62,108,161]. Nevertheless, natural recombination phenomena have been identified in relation to genotype A [153,159]. Samples classified as A9/A11 and A3/A11 were reported by Bazzucchi et al. [143] based on gag-pol region analysis, and a sample defined as A4 based on hypervariable region analysis of env gene was identified by Blatti-Cardinaux et al. [100]. Only one work identified A5/A12 recombination based on LTR sequences, but the results were confirmed by only two out of seven methods tested [10]. Recombination studies have also been conducted on complete genomes classified as A2 [146], allowing for accurate quantification of intra-host genetic diversity. Genotype B is generally more homogenous [149]. However, in Canada, a sample was identified that seems to be the result of a recombination event between B1 and A2 [160]. This result was demonstrated using multiple recombination analysis software. In Poland, a recombinant sample A12/B2 [116] was identified based on the gag gene analysis, using five different statistical methods. Ancestral and recent recombination phenomena based on the analysis of complete genomes have also been hypothesized by Carrozza and colleagues [169].

Few studies among those reviewed focused on the correlation between phylogenetic analysis and compartmentalization. Deubelbeiss and colleagues identified different subgenotypes within the same animal but in different anatomical sites [145]. Blatti-Cardinaux and colleagues, through the study of a clone A4, identified different patterns of pathogenicity depending on the type of infected cell, conducting both in vivo and in vitro studies [100]. An important contribution was made by the work of Pisoni and colleagues, who analyzed the distribution of env sequences in blood and colostrum samples [71]. Based on the results obtained, they demonstrated the compartmentalization of SRLVs between colostrum and blood of infected goats. Finally, Olech and Kuzmak applied six different statistical methods to analyze results obtained from three seropositive goats [115], based on the phylogenetic analysis of env, gag, and LTR sequences. Evidence of compartmentalization was found in all three animals.

## 4. Discussion

One of the most notable findings in this study is the imbalance in the geographical representation of the published sequences, with most coming from Europe and North America, while entire continents are heavily underrepresented, or even completely absent [102,168,170]. This discrepancy is likely due to structural barriers and economic limitations, particularly in developing countries. However, considering the global growth of sheep and goat farming [176], it is crucial to expand research efforts to acquire new valuable insights that would be useful for minimizing economic losses associated with SRLVs.

In the absence of a gold standard, molecular characterization studies are crucial for developing sensitive and specific diagnostic tests, providing updated and reliable epidemiological insights, performing correlation studies between subgenotypes and resistant breeds and reconstructing subgroups movements and SRLV evolution over time [24,26,58,72,95,111,127,128,132,141,146,154,155,166]. Due to the extensive genetic variability of SRLVs, developing different diagnostic protocols tailored to specific genes or genomic regions is advisable. Rapid and reliable diagnostic tests, potentially combining serological screening and molecular analysis, are particularly valuable for timely diagnoses, especially during new introductions, suspected outbreaks, or outbreaks [61,67,149,164]. The optimal combination can be established through the utilization of Composite Reference Standards, which serve as a fixed rule used to make a final diagnosis based on the results of two or more tests, referred to as component tests [177]. This methodology was recently employed within the SRLV domain [178]. This approach is advised in the absence of a gold standard [177,179], but should be applied accurately to avoid possible bias [180]. Since SRLV infections often exhibit no evident signs of disease and infected animals may remain carriers for life, it is crucial to test asymptomatic animals as well [113,130,151]. This is true especially in the case of infection with low pathogenic strains such as genotype E, which is hypothesized to hamper the appearance of symptoms related to CAEV strains in the case of co-infection [54]. Some researchers propose developing specific diagnostic tests tailored to specific geographical areas [98,149,151]. They argue that diagnostic tests based on a single strain may be unrepresentative due to the high variability of SRLVs [40,111]. Therefore, sequencing a large number of samples collected from a defined area is crucial for gathering the information needed to design specific primers for locally prevalent variants.

Conversely, for gaining a deeper understanding of intra- and inter-subgenotype differences and constructing robust phylogenetic trees with high bootstrap values, several authors advocate selecting gene regions of sufficient length; using short sequences may lead to lose information [62,143]. Based on the reviewed literature, sequences ranging from 600 to 800 bp are often associated with higher bootstrap values [61,65,71,72,99,105,109,115,116,119,124,141,149,163,167], indicating that this width range of genetic fragments could be considered reliable for accurate phylogenetic analysis. Among the studies, the *gag* gene has been the most commonly selected gene for genotyping over the years [72,96,99,103,106,109,113,154,160], alongside the *gag-pol* region [60,161] and the *pol* gene [60,118,135]. Finally, the *env* gene has been extensively used for phylogenetic characterization [72,73,101,124,139,156,165,167]. The *env* gene features both conserved and highly variable segments [12], while the *gag* gene and *gag-pol* region are highly conserved, facilitating the detection of single-nucleotide polymorphisms (SNPs) among different strains and the development of optimized molecular tests [64,103,123,130,150]. The *gag* gene is particularly valued for its low intra-variability within groups and the avaliability of numerous reference sequences from online databases [114].

Zanoni’s work remains pivotal as it emphasized the necessity for phylogenetic classification of SRLVs, highlighting its importance in developing effective containment strategies [66]. However, it suffered from limitations in methodology and a lack of extensive information on SRLV phylogenesis. Specifically, the study used short sequences for phylogenetic analysis. Moreover, in 1998, the online availability of complete sequences was limited, and even partial sequences were sparse and often short (200–300 bp). Shah’s subsequent work contrasted with Zanoni’s approach by moving away from the concept of clades, which presents ambiguity and a lack of equidistance between the clusters from which they are formed, and focusing on genotypes and subgenotypes based on phylogenetic and pairwise distance analysis [60]. Shah identified a challenge in meeting all the criteria for defining new subgenotypes recommended in HIV research [181]. This approach required the entire sequencing of representative strains, which should resemble each other but not other subgenotypes, and they should come from at least three unrelated individuals. This criterion was not met by Shah; however, it was applied in defining the reference isolate Volterra belonging to subgenotype B3 [120], suggesting its applicability in the field of SRLVs. On the other hand, the percentage values suggested by Shah are also based directly on the criteria defined in the HIV classification, ensuring the feasibility and the robustness of the method [181]. Adopting methodologies derived from the HIV research area could prevent incorrect or ambiguous classifications.

It is important to note that sequencing the same sample using different genes as targets can lead to ambiguous genetic classifications [61,62,116,160]. A notable example of this ambiguity is given by genotype C, identified through sequencing of the *pol* gene, but categorized as genotype B when sequenced in the *gag-pol* region [62,107]. Nevertheless, using primers designed for sequencing larger regions it is possible to identify genotype C in both *gag* and *gag-pol* regions [113]. The comparison of studies focusing on genotype C serve as just one example of the profound differences in approaches to SRLVs phylogenetic characterization found in the literature. At the same time, the high genetic variability of SRLVs forces researchers to adopt multi-approach methods, because excessive simplification can result in the loss of information, including variant detection. An interesting solution appears to build multiple phylogenetic trees using different algorithms, in order to compare different procedures applied to the same sample and thus evaluate the robustness of the relevant method.

One strategy to identify as many variants as possible is to use degenerate primers, which contain positions where multiple nucleotides can be present in the mixture [60,150,161,166]. Alternatively, researchers have implemented a genotype-specific approach or utilized multiple sets of primers tailored to different genotypes [128]. Another approach involves the selection of primers based on rare samples, such as genotype E, where generic PCRs have shown limited effectiveness in detecting samples from this recently discovered group [155]. Several studies have focused exclusively on more variable sequences, such as *env* and LTR, considering them more suitable for representing the high genetic variability of small ruminant lentiviruses [10,65,67,129,164] or for studying specific aspects such as promoter activity [133]. Interestingly, some strains identified in Romania based on the *gag* gene were not clearly classifiable, likely due to the excessive conservation of this gene. Therefore, it seems to be recommended to switch to a more variable sequence like *env* to capture more phylogenetic differences [3]. However, there may be a certain failure rate of PCRs or also low bootstrap values in phylogenetic trees, likely due to the difficulty of designing efficient primers for such variable sequences [10,40,129].

Other sequences used over time as targets for phylogeny include regulatory sequences such as *vpr-like* (formerly known as tat) [121]. In the field of HIV research, *vpr-like* plays a fundamental role, containing both extremely variable and more conserved traits, making it suitable for studying the evolutionary mechanisms of different strains [182,183]. In contrast, in the case of SRLVs, this approach is still in its infancy and deserves to be explored further.

The optimal approach could be a compromise between using highly conserved sequences and extremely variable ones. This can be achieved by adopting a mixed approach, which involves sequencing both conserved and variable regions of the same sample in parallel and comparing the results to resolve ambiguities [72,105,108,115,116,130,137,163,167].

Since the early 2000s, a new approach to the phylogeny of SRLVs, involving the sequencing of complete or nearly complete genes, has also been applied. In 2006, a molecular characterization study of the genetic variability of the complete *env* gene of a Norwegian isolate confirmed the previously proposed existence of genotype C, which is closely geolocated in Norway [125]. Through the nearly complete sequencing of the *env* gene (about 1800 bp), researchers were able to clearly depict the genetic distance of this strain from reference sequences, as well as confirm the presence of hypervariable regions V1-V5 [73], information that would have been difficult to obtain through partial sequencing. Over the years, various research groups have adopted the approach of complete gene sequencing, considering it a more informative and robust method compared to partial sequencing, and faster and cheaper than NGS. By analyzing the complete *gag* gene, it has been possible to demonstrate co- or superinfections in animals from mixed flocks, an event rarely reported under field conditions [128].

Recently, the classification of SRLVs into five genotypes (A-B-C-D-E) has been questioned [93]. Some publications have suggested that genotype D does not deviate sufficiently from genotype A to justify its definition as a separated group and should be considered a new A subgenotype [12,63]. Consequently, an updated SRLV classification, particularly focusing on doubtful cases that depend on the phylogenetic method or gene region selected, has been recommended [143,146], as advocated also by Colitti and colleagues. In 2019, they sequenced the complete genomes of 22 new SRLVs strains, conducting a thorough comparison of their genetic properties and in vitro characteristics. They suggest that a similar approach should be adopted during reclassification to achieve a more accurate and updated classification [85]. Conversely, Michiels and colleagues hypothesized that strains identified on short gene fragments provide limited genetic information and may not be sufficiently robust when sequenced using the protocol adopted by Shah et al. [62]. For example, variants found in Brazil based on *pol* gene sequencing have been identified as C-like [135]. However, the phylogenetic analysis was carried out on a sequence about 238 bp long, so it would be appropriate to consider larger regions to obtain more robust data.

A major breakthrough in the molecular characterization of SRLVs has been the sequencing of complete genomes. To date, most research has relied on the overlapping fragments approach [23,54,96,100,117,120,125,142]. However, more recently, the NGS approach has been employed, a transition suggested by several studies [93,108,143]. The initial work in this field was conducted in the United States by Workman and colleagues [171,172] followed by Dickey et al. [146]. Subsequently, the NGS method has also been applied to SRLV research in Europe, further advancing our understanding of the viruses’ genetic diversity and evolutionary patterns.

Currently (August 2024), online databases contain fewer than 100 complete genomes (database GenBank^®^, NIH), with 22 of these contributed by Colitti [85]. This represents a valuable contribution, reinforcing the belief that complete genomes provide critical information for developing recombinant antigens necessary for sensitive and reliable serological screening tests, as well as designing primers for robust molecular diagnostics. The need to expand these databases with complete genomes is supported by various studies [146,171,172], which suggest taking inspiration from the characterization of circulating recombinant forms (CRFs) in HIV research, emphasizing the importance of creating unique reference models for SRLV genotyping [146]. In recent years, several NGS studies have been published in China, addressing the lack of information and a lack of understanding of the local epidemiological landscape. These studies confirm the importance of characterization for providing valuable support for eradication and containment plans, as well as the need for continents other than Europe and North America to sequence and make available acquired information. Previous difficulties in identifying Chinese SRLVs may have been caused also by the use of primers designed on reference genomes that were excessively variable compared to the strains that are present in China. Therefore, sequencing strains from various countries are desirable [168]. Finally, NGS enables studies on the genetic evolution of SRLVs [86,138] and the identification of common ancestors [170], which is crucial for understanding viral movements and genetic flows.

Phylogenetic analyses of sequenced samples allow the collection of in-depth information, for example, to distinguish samples derived from recombination phenomena from natural mutants [184,185], a proven concept in the field of HIV [181]. Natural recombination in SRLVs can lead to the emergence of subgenotypes with intermediate characteristics between different strains, which over time can form distinct subgenotypes. This process is often attributed to coinfections in the same animal with multiple strains, as suggested before [12,116,146]. However, this does not always occur, even in conditions of double-infection [161]. Although research on this issue is limited, recombination in nature plays a significant role in increasing SRLV variability and forming new variants that can evade standard diagnostic tests used in specific geographical area [143,153,159]. Addressing recombination studies can help resolve ambiguities in previous phylogenetic classification [100,160]. These studies may also demonstrate that samples that are classifiable into different subgeotypes may not constitute clearly separated groups but are recombinants [116,146]. Generally, the hypervariable *env* region appears particularly informative for recombination studies, whereas data for the LTR region are too sparse to assess its suitability, necessitating further research.

Finally, phylogenetic evidence of the ancient separation of genotypes A and B was identified using the Lentivirus-GLUE resource, which is a framework that enables the re-use of genomics datasets across diverse analysis contexts for comparative genomic analysis of lentiviruses, developed using the GLUE software framework [169]. This very recent approach has made it possible to hypothesize an Eastern origin for genotype A and a Western origin for genotype B, showing a correlation between the spread of certain sheep and goat species and the viruses of both groups. This relationship also seems to influence the level of pathogenicity of certain subgenotypes once they are introduced into breeds with which they have no long-term associations. It might be interesting to pursue this type of study because it could provide useful tools for the control/eradication of SRLVs.

A field that remains relatively unexplored and is worthy of further study is the compartmentalization of SRLVs, sometimes only hypothesized rather than proven [3]. Working with optimal matrices yields more robust data. Therefore, the collection of representative samples is crucial. To this end, expanding research on compartmentalization is essential to avoid samples with proviral loads that are too low [71,145]. Such studies could provide additional insights into the pathogenicity or replication efficiency of specific variants, which would be useful for developing targeted diagnostic tests [100]. Moreover, these studies could help clarify the role of certain genes in the cellular tropism of SRLVs, a mechanism that remains unclear to date [115]. Analyzing the distribution of SRLVs in different tissues has shown that they predominantly replicate in the mammary glands, significantly increasing the likelihood of transmission to offspring through lactation [71]. This factor should be considered in control and prevention plans for flocks. We recommend deepening this issue with dedicated studies.

## 5. Conclusions

In an increasingly globalized world interconnected by commercial networks, limiting the transmission of SRLVs is a critical goal. Achieving this requires expanding our understanding of different epidemiological situations through increased sequencing efforts, particularly in underrepresented countries. Similarly, more research on SRLVs in wild animals may also enhance our ability to establish comprehensive epidemiological profiles, shedding light on potential reservoirs and transmission pathways. Understanding these dynamics will be key to devising effective control and prevention strategies for SRLVs. Additionally, adopting a statistical approach is crucial to ensure that a representative number of samples are collected and analyzed.

Due to the high genetic variability of SRLVs, achieving the standardization of tests used for molecular characterization is desirable to avoid the excessive proliferation of subgenotypes based on weak and unreliable data. Concurrently, a serological screening approach should be pursued. We recommend developing a diagnostic procedure based on the Composite Reference Standard method to achieve the most accurate diagnoses possible.

Knowing which strains are present in a given area is our main defense against SRLVs in the absence of vaccinations and treatments. For phylogenetic studies, NGS or whole gene sequencing is recommended whenever possible. If these methods are not feasible, using wide sequences is advisable to obtain robust data. Based on our results, we suggest sequencing genomic regions ranging from 600 to 800 bp. This approach could be particularly important as a review of the systematic classification of SRLVs is urgently needed, especially given the ambiguities surrounding genotypes C and D. Expanding our knowledge of SRLV variability could also benefit from sequencing the *vpr-like* regulatory gene, which appears promising. Additionally, more attention and resources should be allocated to studies on recombination and compartmentalization, as they provide valuable insights into the evolutionary dynamics and adaptation mechanisms of SRLVs. Finally, given that the applicability of HIV-derived methods has already shown promise in SRLV molecular characterization, we propose their broader implementation. Applying these methods on a large scale for the phylogenetic classification of SRLVs, particularly in terms of sample size, statistical approaches, and criteria for defining new subgenotypes, could significantly enhance the robustness and consistency of SRLV research and diagnostics.

## Figures and Tables

**Figure 1 animals-14-03545-f001:**
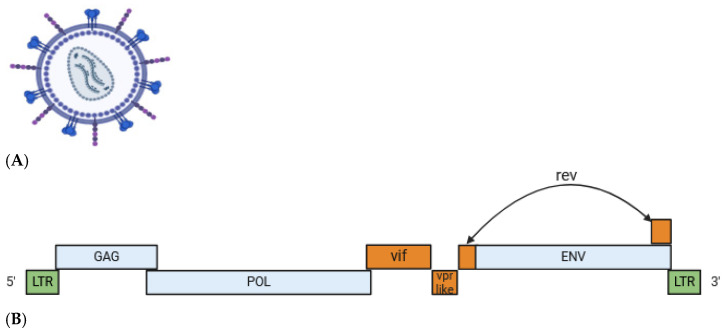
(**A**) SRLV viral particle, containing two copies of (+) ssRNA. (**B**) Schematic representation of the SRLV genome organization. Structural genes are shown in blue, flanking sequences in green, and regulatory genes in orange (created by Biorender.com, accessed on 17 June 2024).

**Figure 2 animals-14-03545-f002:**
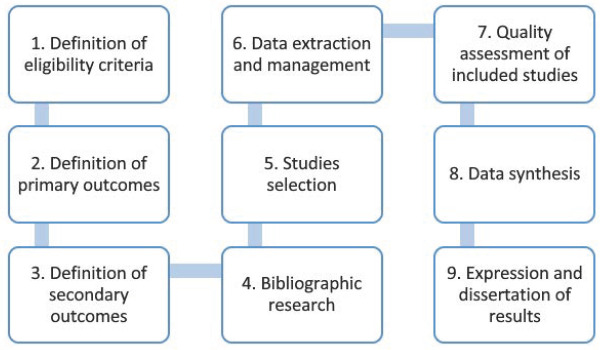
Schematic representation of the Materials and Methods.

**Figure 3 animals-14-03545-f003:**
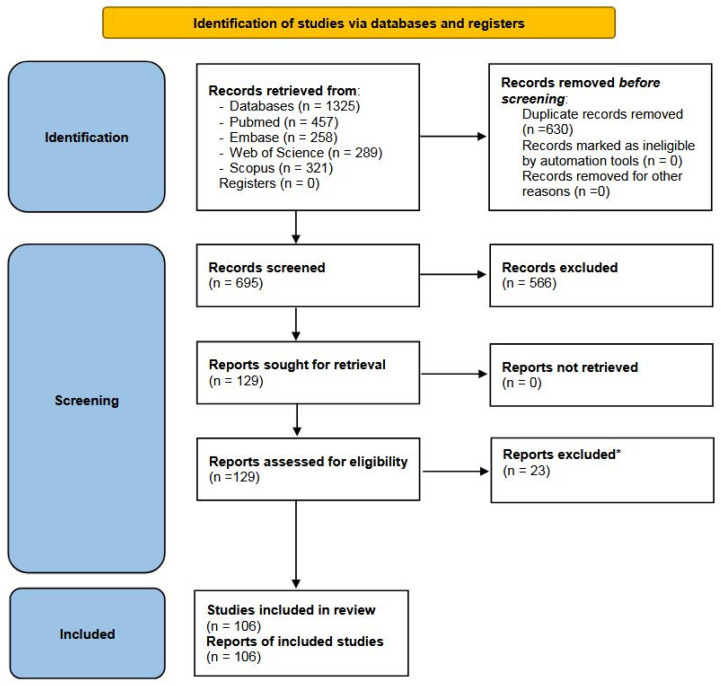
PRISMA 2020 flow diagram for systematic reviews, which only includes searches of databases and registers [90]. * As reported in Appendix A.

**Figure 4 animals-14-03545-f004:**
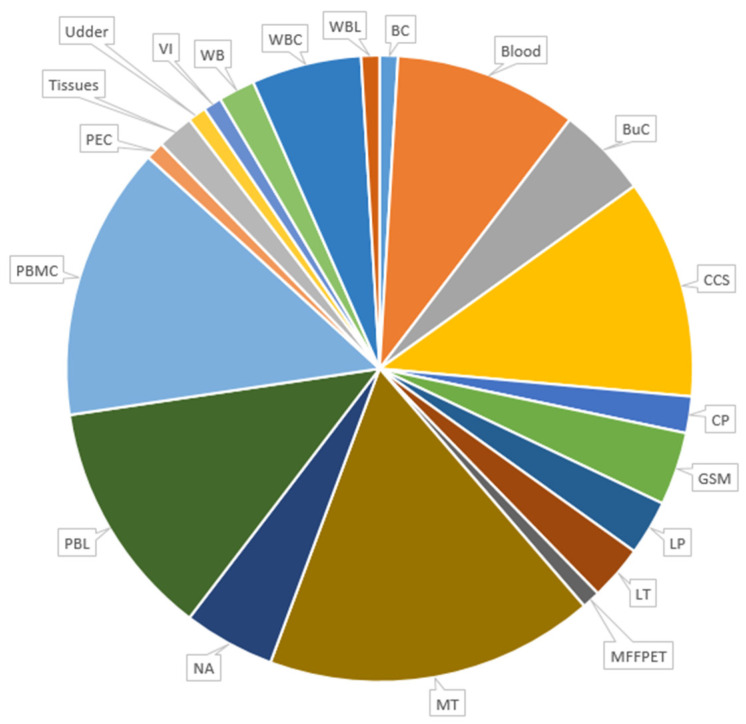
Types and number of samples analyzed in the selected papers for this systematic review. BC: blood clot; BuC: buffy coat; CCS: cell culture supernatant; CP: cell pellet; GSM: goat synovial membrane; LP: leukocyte pellet; LT: lung tissue; MFFPET: multiple formalin-fixed paraffin embedded tissue; MTs: multiple types (when different types of samples have been examined); NA: data not available; PBL: peripheral blood leukocyte; PBMC: peripheral blood mononuclear cell; PEC: primary epididymus culture; VI: virus isolate; WBC: white blood cell; WBL: white blood leukocyte; WB: whole blood.

**Figure 5 animals-14-03545-f005:**
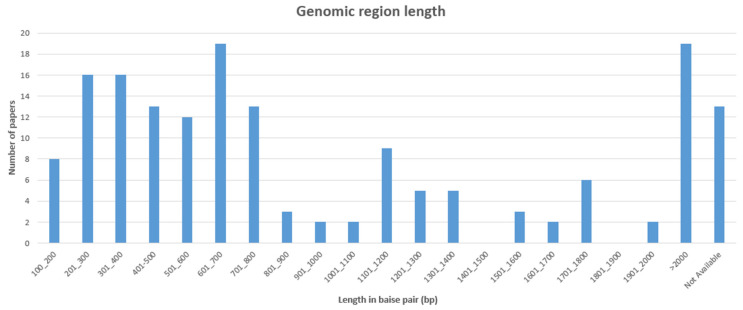
Bar chart showing length of genomic regions used for subgenotyping.

**Figure 6 animals-14-03545-f006:**
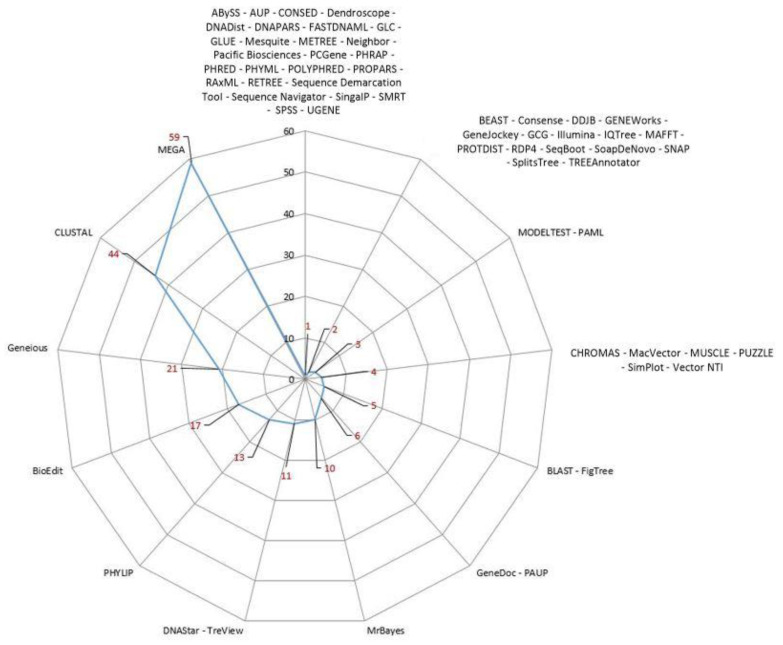
Radar chart illustrating the software used for the phylogenetic analyses in the retrieved papers. Each point on the radar chart represents a different software program, and the number of papers using each software is reported in red.

**Figure 7 animals-14-03545-f007:**
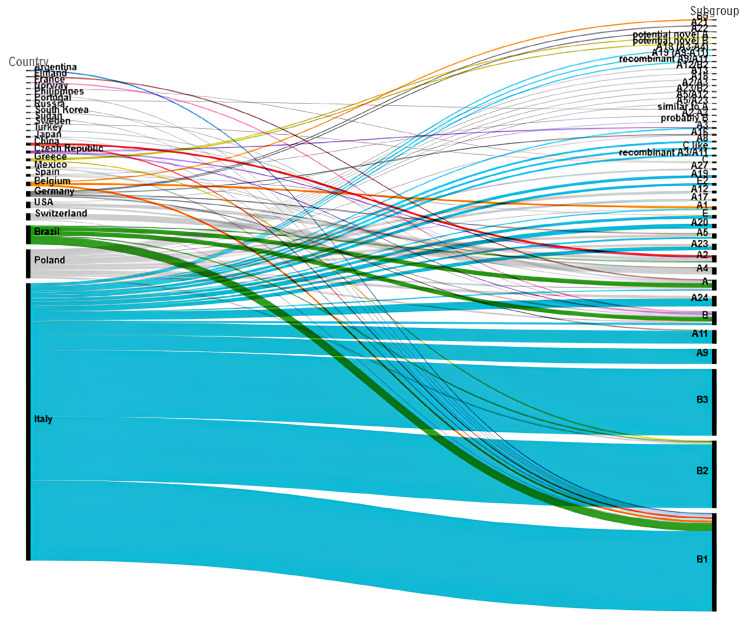
Alluvial plot of the global distribution of SRLV subgenotypes, based on the current SRLV classification, created by Rawgraphs.io (https://www.rawgraphs.io, accessed on 06 December 2024). The plot visually represents the relationship and distribution patterns of various SRLV subgenotypes across different geographic regions, without considering the potential misclassification of some subgenotypes.

**Table 1 animals-14-03545-t001:** Quality assessment and risk of bias results: the evaluated papers are categorized by article type and quality assessment outcomes.

Type of Paper	Evaluation ^1^	Number of Evaluated Paper
Case report	≤60%	0
	from >60% to ≤80%	2
	>80%	1
Cross-sectional studies	≤60%	9
	from >60% to ≤80%	41
	>80%	51

^1^: ≤60% unsatisfactory; from >60% to ≤80% good; >80% very good.

**Table 2 animals-14-03545-t002:** Quality assessment of narrative reviews based on SANRA tool [92].

Paper Evaluated	SANRA Evaluation
Ramirez et al. 2013 [12]	7/12
Olech 2023 [93]	9/12

## Data Availability

No new data were created or analyzed in this study. Data sharing is not applicable to this article.

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
