# Peer review of "Molecular Characterization of Small Ruminant Lentiviruses in Sheep and Goats: A Systematic Review"

_animals, 2024, doi:10.3390/ani14233545_

Round 1

Reviewer 1 Report

Comments and Suggestions for Authors

Analyzing the article on Small Ruminant Lentiviruses (SRLVs) reveals a blend of ambition and underlying uncertainty. The authors strive to tackle the critical and economically impactful challenge of SRLV genetic diversity and global diagnostic efforts. Yet, upon closer examination, it becomes evident that the study lacks the robust foundation necessary to substantiate its extensive claims, leaving the overall presentation feeling unfinished and insufficiently developed. The introduction quickly captures the reader's interest by emphasizing the need to understand SRLVs due to their effect on the health and productivity of small ruminants. However, the narrative shifts into broad and sweeping statements. The review claims to address an extensive range of topics—from molecular detection techniques to software for phylogenetic analysis—without presenting a clear and structured methodology. Phrases like "comprehensive overview" are used but remain unexplained, raising questions about whether the review truly offers novel insights or merely reiterates existing knowledge. The authors advocate for substantial measures, such as enhancing sequencing initiatives globally, particularly in areas with limited data. Nevertheless, they overlook critical barriers, including logistical and financial constraints, that would complicate the implementation of these suggestions. Simple statements, like "limiting the transmission of SRLVs would be desirable," appear overly reductive, disregarding the complex realities of disease management in livestock populations. Additional concerns arise from the suggestion to standardize molecular diagnostic tests to curb the creation of unreliable subgenotypes. While this idea is conceptually valid, the conclusions lack a detailed plan or discussion on how such standardization could be practically achieved or funded. Recommendations like establishing "a diagnostic procedure based on cross-referenced standards" seem appealing but are presented without sufficient context or feasibility analysis. Ultimately, while the article successfully identifies the pressing need for better SRLV diagnostic and classification methods, it falls short of providing a concrete, actionable strategy. The recommendations are broad and, at times, vague, lacking practical or evidence-based solutions.

Reviewer 2 Report

Comments and Suggestions for Authors

The authors of this excellent paper performed  systematic literature review of small ruminant lentiviruses circulating in different part of the world. This is the first study of its kind to focus the reader's attention on the molecular characteristics of SRLVs,  including country-specific data.

This study systematically searched Medline, Embase, Scopus and Web of Science for all empirical molecular  studies that presented SRLV genotype - specific data from all countries of the world where SRLV infections were detected. SANDRA tool was used to assess the quality of included studies. Overall, 1,325 published articles were retrieved from these four databases

It is important that in this analysis the authors focused on the relevant elements associated with SRLV infections like: geographical origin of samples and animal species, genome fragments selected for genotyping, diagnosis of infection with SRLV, phylogenetic analysis, recombinant events and compartmentalization   In section 3.2, lines 306-308, the authors present data from sheep, goats or both species; however, they do not mention wildlife ruminants.  Whence it is known that data from such animals have been described, at least in terms of the occurrence of particular genotypes.  

Certainly, this is a relevant contribution to the field of infection with SRLV and the work is within the scope and interests of the VIRUSES and the paper should be published.

Reviewer 3 Report

Comments and Suggestions for Authors

The paper presents a systematic review of approaches used to molecularly characterize small ruminant lentiviruses (SRLVs). While the study is timely and has the potential to offer a valuable synthesis, it faces several structural and clarity issues. The paper’s organization appears to alternate between a traditional review format and a systematic review, creating an unclear identity for the reader.

The text occasionally lacks focus, with sections in the discussion that might be more suited to the results, and introductory material that could benefit from greater emphasis on framing the purposes of the systematic review. The introduction, while thorough on SRLV genetics, would benefit from a clearer explanation of the study’s specific aims within the context of the review.

Some discussion points, such as the transition to next-generation sequencing (NGS) methods, may appear as general observations rather than findings from the systematic review. Overall, there is some ambiguity between conclusions drawn directly from the systematic review—such as tissue-source trends across studies—and interpretations synthesized from prior literature.

For clarity, I suggest revisiting the paper’s organization to deliver a focused message, potentially on developing community standards. Recognizing the different contexts of SRLV sequence analysis could clarify the review’s focus, distinguishing between operational applications (e.g., diagnostics, genotyping) and more specialized research contexts (e.g., compartmentalization studies).

With respect to the findings of systematic review, there could perhaps be a more concise discussion. The main message seems to be that a great diversity of different approaches have been used historically, and there is a need for new standards that are up to speed with modern sequencing techniques. The sequencing studies so far performed for SRLVs are fairly heterogeneous in nature and there may not be much more to say about it than that.

The analysis of software usage is somewhat unclear in its value, as it appears to conflate distinct types of sequence-based analyses. I was intrigued by Figure 6 but not entirely sure what it was telling me.

The legacy subtype nomenclature based on subgenomic sequences is noted as being limited in informativeness. Consequently, Figure 7, while visually appealing, may require additional context clarifying that much of the existing nomenclature, established on subgenomic sequences, might require revision in light of whole-genome relationships.

Minor Edits

    p17, line 163: "researcherers" should be "researchers."
    p16, line 571: "builded" should be "built."
    p18, lines 700-708: My understanding of this from reading the website is that the GLUE framework enables the re-use of genomics datasets across diverse analysis contexts. The SRLV extension of Lentivirus-GLUE exemplifies this, with a comprehensive SRLV dataset initially used to explore viral origins subsequently repurposed for genotyping.
